# Sexual Dysfunction and Quality of Life in Patients with Hidradenitis Suppurativa and Their Partners

**DOI:** 10.3390/ijerph20010389

**Published:** 2022-12-26

**Authors:** Carlos Cuenca-Barrales, Trinidad Montero-Vilchez, Piotr K Krajewski, Jacek C Szepietowski, Lukasz Matusiak, Salvador Arias-Santiago, Alejandro Molina-Leyva

**Affiliations:** 1Hidradenitis Suppurativa Clinic, Dermatology Department, Hospital Universitario Virgen de las Nieves, 18014 Granada, Spain; 2TECe19-Dermatología Clínica y Traslacional Investigation Group, Instituto de Investigación Biosanitaria ibs.GRANADA, 18012 Granada, Spain; 3Department of Dermatology, Venereology and Allergology, Wroclaw Medical University, 50-368 Wroclaw, Poland; 4School of Medicine, Department of Dermatology, University of Granada, 18016 Granada, Spain

**Keywords:** hidradenitis suppurativa, quality of life, sexual dysfunction, erectile dysfunction, sexual partners

## Abstract

Hidradenitis suppurativa (HS) is a chronic skin disease that profoundly affects patients’ quality of life (QoL) and sexuality. Few data are available on the impact of HS on patients’ partners. We aimed to explore the QoL and sexual function of patients’ partners and the factors associated with their impairment and to compare the QoL and sexual function of single patients and those in a stable relationship. We conducted a cross-sectional study at Hospital Universitario Virgen de las Nieves (Granada, Spain) and at the Wroclaw Medical University (Wroclaw, Poland). Patients over the age of 16 years attending their scheduled follow-up and their partners, if any, were included. The Dermatology Life Quality Index (DLQI) and the Family Dermatology Life Quality Index (FDLQI) were used to estimate the QoL of patients and partners, respectively. The FSFI-6 was used to determine sexual dysfunction in women and the IIEF-5 for erectile dysfunction in men. Thirty-four single patients and twenty-eight patients in a stable relationship and their partners were included. Patients had a very large impact (DLQI 12.5 ± 7.5) and their partners a moderate impact (FDLQI 10.3 ± 7.1) in their QoL due to HS. Women with HS had a high prevalence of sexual dysfunction (13/32, 40.6%) and men of erectile dysfunction (19/30, 63.3%). Male partners also had a high prevalence of erectile dysfunction (10/17, 58.8%), while only one female partner had sexual dysfunction (1/11, 9.1%). Factors related to disease severity, intensity of symptoms and body mass index were associated with poorer QoL in partners and time of disease evolution with greater erectile dysfunction in male partners. In conclusion, HS not only profoundly affects the QoL and sexuality of patients but also of their partners. Several risk factors have been identified, which should be taken into account in the holistic approach of the disease.

## 1. Introduction

Hidradenitis suppurativa (HS) is a chronic, inflammatory, recurrent, debilitating skin disease of the hair follicle that usually presents after puberty with painful, deep-seated, inflamed lesions in the apocrine gland-bearing areas of the body [1]. HS has an estimated prevalence of about 1% in Western countries [2]. When the disease is uncontrolled, it may progress causing tissue damage, with scarring and fistulae, and uncomfortable symptoms such as pain, suppuration, unpleasant odor and pruritus [3,4]. The etiopathogeneses of HS seems to be multifactorial, but this is not completely understood [5,6,7]. HS was originally only regarded as a cutaneous disorder, but today, a growing body of evidence links HS with several dermatological and nondermatological comorbidities, such as metabolic syndrome, inflammatory bowel disease, cardiovascular risk, depression or anxiety, among others [8,9,10,11,12]. The management of this disease is a combination of medical therapies and surgical techniques [13,14].

It is therefore to be expected that the disease will affect patients, but also their partners, as occurs in other dermatoses [15]. The profound impact of HS on patients’ quality of life (QoL) has been well documented [16,17,18,19], being greater than that of other dermatoses and comparable to diseases such as chronic obstructive pulmonary disease, cardiovascular disease, cancer or diabetes [20,21]. The interference of the disease with sexual functioning of HS patients has also been investigated, with a prevalence of sexual dysfunction in women ranging from 51% to 62% and of erectile dysfunction in men ranging from 52% to 60% [22]. Psychological comorbidities, including depression and low self-esteem, as well as metabolic and cardiovascular diseases, could also impair sexual function in patients with HS [23]. Moreover, unhealthy lifestyles—including smoking, alcohol, substance abuse and chronic stress—also contribute to the development sexual dysfunctions [24].

Several risk factors were identified, with the presence of a stable partner being an important preventive factor for sexual dysfunction, sexual distress and feelings of fear of rejection or low perceived attractiveness [25,26,27]. However, there is no research on the impact of HS on the sexuality of patients’ partners and only one study on its impact on their QoL [28]. The aim of this research was to explore the QoL and sexual function of patients’ partners and the factors associated with their impairment and to compare the QoL and sexual function of single patients and those in a stable relationship.

## 2. Materials and Methods

### 2.1. Design

We conducted a cross-sectional study from January to September 2020. Patients were recruited at the Hospital Universitario Virgen de las Nieves in Granada (Spain) and at the Wroclaw Medical University in Wroclaw (Poland) in dermatology departments specialized in HS management.

### 2.2. Inclusion and Exclusion Criteria

Patients over the age of 16 years attending their scheduled check-ups in both departments were invited to participate in the study. Partners of patients in a stable relationship were also invited to participate. Patients with intellectual disabilities, those with diagnosed psychiatric comorbidities or those who did not consent to participate in the study were excluded.

### 2.3. Ethics

The study was approved by the Institutional Review Board of Hospital Universitario Virgen de las Nieves and is in accordance with the Declaration of Helsinki. Patients were aware of the anonymous treatment of their data and gave their informed consent to participate in the study.

### 2.4. Variables of Interest

Clinical, biometric and socio-demographical variables, including whether the patient was single or in a stable relationship, were recorded by means of clinical interview and physical examination. Socio-demographical information from their partners (if any) was also recorded.

Structural damage was assessed by means of the Hurley staging system: stage I (abscess formation, single or multiple, without sinus tracts or scarring), stage II (recurrent abscesses with tract formation and scarring, single or multiple, widely separated lesions) and stage III (diffuse or near-diffuse involvement or multiple interconnected tracts and abscesses across the entire area).

Inflammatory load was assessed using the International Hidradenitis Suppurativa Severity Score System (IHS4). The formula for calculating the IHS4 is as follows: the number of inflammatory nodules multiplied by 1 plus the number of abscesses multiplied by 2 plus the number of draining fistulae multiplied by 4. Results range from 0 to infinite. Scores under 4 indicate a mild inflammatory load; from 4 to 10, a moderate inflammatory load; and over 10, a severe inflammatory load [29].

The intensity of symptoms was assessed by means of Numeric Rating Scales (NRS), where the patient noted the intensity of the symptom using a numeric scale graded from 0 (absence of symptoms) to 10 (the highest intensity of symptoms) [30].

The Dermatology Life Quality Index (DLQI) and the Family Dermatology Life Quality Index (FDLQI) were used to assess QoL impairment caused by HS in patients and their partners (if any), respectively. Scores range from 0 to 30; scores between 11–20 indicate a very large impact on the patient’s/partner’s QoL, and above 20 represent an extremely large impact on the patient’s/partner’s QoL [31,32].

The presence of sexual dysfunction in both female patients and partners was evaluated using the Female Sexual Function Index-6 (FSFI-6), a validated questionnaire that explores the six domains of female sexual function (desire, excitation, lubrication, orgasm, global satisfaction and pain), each one with a single question. A score of 19 or less indicates sexual dysfunction with 96,1% of sensitivity and 90,9% of specificity [33]. The presence of erectile dysfunction in both male patients and partners was assessed by means of the International Index of Erectile Function-5 (IIEF-5), a validated questionnaire with 5 questions about erectile function. A score of 21 or less indicates erectile dysfunction (ED) with 98% of sensitivity and 88% of specificity.

### 2.5. Statistical Analyses

Variables’ normality was explored by means of the Shapiro–Wilk test and histograms. Descriptive statistics were used to explore the characteristics of the sample. Continuous variables were expressed as means and standard deviation. Qualitative variables were expressed as absolute and relative frequencies. To explore factors possibly associated with FDLQI, FSFI-6 and IIEF-5 scores in patients’ partners, Student’s *t*-test was used for dichotomous variables, one-way analysis of variance for nominal variables with two or more categories and simple linear regression for continuous variables. Significance was set for all tests at two tails, *p* < 0.05. Statistical analyses were performed using JMP version 14.1.0 (SAS Institute, Cary, NC, USA).

## 3. Results

### 3.1. Sociodemographic Characteristics

A total of 34 single patients (15 women and 19 men) and 28 patients in a stable relationship (17 women and 11 men) and their partners were included. No participant refused to enroll in the study. All couples were heterosexual. The mean age of patients with HS was 36.8 (11.9), and that of their partners was 42.4 (10.8). There were statistically significant differences in age between patients with and without a stable relationship. There were no differences regarding sex distribution or residence country. Body mass index (BMI) was significantly higher in patients in a stable relationship (Table 1).

### 3.2. Disease Characteristics

Disease duration was longer in patients in a stable relationship compared to single patients. The severity of the disease, in terms of Hurley stages, and the inflammatory load, measured using the IHS4, were similar in both groups. Regarding disease distribution, axillae were more commonly affected in single patients; there were no differences in other areas. The intensity of symptoms was also comparable between both groups (Table 1).

### 3.3. Quality of Life and Sexual Function in Patients with HS

The mean DLQI score (12.5 ± 7.5 points, ranging from 0 to 29) indicated a very large impact of HS on patients’ QoL. More than 90% (31/34) of single patients and 78.5% (22/28) of patients in a stable relationship experienced a moderate, very large or extremely large impact on QoL due to HS.

The mean FSFI-6 and IIEF-5 scores in both single patients and those in a stable relationship ranged between 18 and 18.9, that is, in the range of sexual/erectile dysfunction. After applying the cut-off points—indicating sexual dysfunction in women as a score of 19 or less FSFI-6 and ED in men as a score of 21 or less in IIEF-5—40.6% (13/32) of women and 63.3% (19/30) of men with HS had sexual and erectile dysfunction, respectively, with no differences regarding the presence of a stable relationship (Table 2).

### 3.4. Quality of Life and Sexual Function in Patients’ Partners

The mean FDLQI score (10.3 ± 7.1 points, ranging from 1 to 29) indicated a moderate impact of HS on partners’ QoL: 74.9% (21/28) of partners reported a moderate, very large or extremely large impact on their QoL due to HS, comparable to data from patients in a stable relationship. There were no significant differences between men and female scores (9.3 ± 2.6 vs. 10.9 ± 1.7 points, respectively, *p* = 0.57).

The mean FSFI-6 score was not in the range of sexual dysfunction; after applying the cut-off point, only one (9.1%) female partner had sexual dysfunction. Neither was the mean IIEF-5 score in the range of sexual dysfunction; after applying the cut-off point, 58.8% (10/17) of male partners had erectile dysfunction (Table 3).

### 3.5. Factors Associated with Lower FDLQI, FSFI-6 and IIEF-5 Scores in Patients’ Partners

Factors associated with poorer QoL and sexual function in patients’ partners were also explored (Table 4). We observed a positive correlation between patients’ BMI, Hurley stage, IHS4 score, NRS for unpleasant odor and NRS for pruritus and FDLQI scores and between longer disease duration and IIEF-5 scores in male partners. We did not find any factors significantly associated with lower FSFI-6 scores in female partners.

## 4. Discussion

In this multicentric cross-sectional study, we have studied the impact of HS on patients’ partners in terms of QoL and sexual function. We have identified several disease-related factors that are associated with poorer QoL for the patients’ partners. We have also investigated the importance of partners for HS patients, comparing QoL and sexual function between single patients and those in a stable relationship.

HS, due to its relapsing and unpredictable course and its unpleasant symptoms, has a variety of personal and social consequences for patients. The QoL of HS patients is impaired to a greater extent than in other dermatological diseases [20]. We observed that the mean DLQI score was in the range of very large impact on patients’ QoL, as previously reported [34]. HS patients are affected by anxiety, depression, stigmatization, low self-esteem, loneliness, unemployment and even increased suicide risk [34,35,36,37,38,39]. In addition, there is a high prevalence of sexual and erectile dysfunction among HS patients, and they suffer from high levels of sexual distress; these disorders are mainly related to the severity and distribution of the disease, the intensity of symptoms and the absence of a stable partner [22]. In our investigation, we observed a slightly lower prevalence of sexual dysfunction in women than previously reported and a similar prevalence of erectile dysfunction in men. However, we found no statistically significant differences in sexual function in women and erectile function in men between single patients and those in a stable relationship, probably due to the smaller sample size compared to previous research [9] and age difference between both groups, as aging is associated with sexual and erectile dysfunction, and patients in a stable relationship were older in our sample.

Despite these findings, the importance of partners for patients with chronic diseases is known, as they give them emotional support. This is especially relevant in diseases with body image disturbances such as HS [40], as partners may promote positive body images [41]. Nevertheless, no attention is paid to the fact that patients’ partners may also be affected emotionally and in various spheres of their QoL by the patient’s disease. There are several investigations on the impact of dermatological diseases on cohabitants’ patients, such as psoriasis [15], HS [42], atopic dermatitis [43], vitiligo [44], epidermolysis bullosa [45] or leg ulcer [46]. However, little is known about the impact of dermatological diseases on patients’ partners, as there is little research specifically focused on them.

We have observed that HS causes a moderate, very large or extremely large impact on three out of four patients’ partners; these figures are higher than previously reported [28] and show that HS has a significant and unrecognized effect not only on patients but also on their partners. As previously reported [28], disease severity, measured in terms of Hurley stages and IHS4 score, was associated with worse QoL in patients’ partners, but we also identified new risk factors, such as patients’ higher BMI and the intensity of unpleasant odor and pruritus. These two symptoms have been described as key symptoms in patients’ QoL impairment [4], but for the first time, their impact on patients’ partners has been identified. Unpleasant odor is an important symptom for HS patients because, unlike other symptoms, it can be perceived by others. In fact, as patients may get used to it [4], it may become even more problematic for their partners than for the patients themselves. On the other hand, pruritus can cause anxiety, interfere with daily activities and decrease sleep quality [47,48], which may also affect patients’ partners.

As for sexual and erectile dysfunction, this is the first study to explore these in HS patients’ partners. We have observed a high prevalence of erectile dysfunction in male partners, greater than that of sexual dysfunction in their female partners with HS and similar to that reported in men with HS [22]. This finding illustrates the profound consequences of HS on the sexuality of patients’ male partners and should be interpreted in light of the importance of partners in the sexuality of women with HS [25,26,27]. A longer time of disease evolution was associated with greater erectile dysfunction in male partners, which may indicate that the disease over time is a strain on sexual relations for the couple. Among female partners, the prevalence of sexual dysfunction was lower than that of erectile dysfunction in their male partners with HS and that reported for women with HS [22]. No factors associated with sexual dysfunction in female partners were identified.

The main limitation of our research is the relatively small sample size and the age difference between single patients and those in a stable relationship, which may have been the reason for the lack of statistically significant differences in QoL and sexual function between both groups. In addition, both study centers specialize in HS, so cases may be more severe than usual.

## 5. Conclusions

In conclusion, HS not only profoundly affects the QoL and sexuality of patients but also those of their partners. Modifiable factors, such as the severity and inflammatory load of the disease, the intensity of unpleasant symptoms such as malodor and pruritus, or BMI, influence this impact. Given the importance that partners seem to have for HS patients, it is essential for the dermatologist to take these facts into account when approaching the disease in order to provide a more holistic approach to treatment and to avoid deterioration of relationships, making it more likely that patients will keep their partners.

## Figures and Tables

**Table 1 ijerph-20-00389-t001:** Sociodemographic and disease characteristics of HS patients.

	Single Patients (*n* = 34)	Patients in a Stable Relationship (*n* = 28)	*p* Value
Age (years)	33.1 (1.9)	41.3 (2.1)	**<0.01**
Sex			
Male	19/34 (55.9%)	11/28 (39.3%)	0.21
Female	15/34 (44.1%)	17/28 (60.7%)	
Residence country			0.61
Spain	18/34 (52.9%)	17/28 (60.7%)	
Poland	16/34 (47.1%)	11/28 (39.3%)	
BMI	28.5 (1.1)	31.9 (1.2)	**<0.05**
Disease duration (years)	9.3 (1.8)	14.7 (2)	**<0.05**
Hurley stage			0.58
I	9/34 (26.5%)	6/28 (21.4%)	
II	21/34 (61.7%)	16/28 (57.2%)	
III	4/34 (11.8%)	6/28 (21.4%)	
IHS4	11.2 (1.7)	11.7 (1.9)	0.85
Locations			
Axilla	26/34 (76.5%)	10/28 (35.7%)	**<0.01**
Breast	6/34 (17.7%)	3/28 (10.7%)	0.5
Groin	17/34 (50%)	20/28 (71.4%)	0.08
Genitals	8/34 (23.5%)	9/28 (32.1%)	0.57
Buttocks	6/34 (17.7%)	6/28 (21.4%)	0.76
NRS for pain	5.1 (0.6)	4.4 (0.7)	0.48
NRS for pruritus	3.1 (0.6)	3.5 (0.6)	0.61
NRS for unpleasant odor	3.6 (0.6)	4.3 (0.7)	0.47
NRS for suppuration	4.2 (0.6)	4.5 (0.7)	0.75

Continuous variables are expressed as means (standard deviation) and qualitative variables as absolute (relative) frequencies. Statistically significant values (*p* < 0.05) are in bold. BMI: body mass index. IHS4: International Hidradenitis Suppurativa Severity Score System. NRS: Numeric Rating Scale.

**Table 2 ijerph-20-00389-t002:** Quality of life and sexual function in patients with HS.

	Single Patients (*n* = 34)	Patients in a Stable Relationship (*n* = 28)	*p* Value
DLQI (score)	11.9 (1.3)	13.3 (1.4)	0.49
DLQI (categories)			0.18
No effect at all on patient’s life	1/34 (2.9%)	1/28 (3.6%)	
Small effect on patient’s life	2/34 (5.9%)	5/28 (17.9%)	
Moderate effect on patient’s life	17/34 (50%)	6/28 (21.4%)	
Very large effect on patient’s life	8/34 (23.5%)	10/28 (35.7%)	
Extremely large effect on patient’s life	6/34 (17.7%)	6/28 (21.4%)	
FSFI-6 (score)	18 (1.9)	18.1 (1.8)	0.97
Sexual dysfunction in women with HS (yes)	7/15 (46.7%)	6/17 (35.3%)	0.51
IIEF-5 (score)	18 (1.2)	18.9 (1.6)	0.64
Erectile dysfunction in men with HS (yes)	12/19 (63.2%)	7/11 (63.6%)	0.98

Continuous variables are expressed as means (standard deviation) and qualitative variables as absolute (relative) frequencies. DLQI: Dermatology Life Quality Index. FSFI-6: Female Sexual Function Index-6. IIEF-5: International Index of Erectile Function-5.

**Table 3 ijerph-20-00389-t003:** Quality of life and sexual function in patients’ partners.

	Partners of HS Patients in a Stable Relationship (*n* = 28)
FDLQI (score)	10.3 (1.3)
FDLQI (categories)	
No effect at all on partners’ life	1/28 (3.6%)
Small effect on partners’ life	6/28 (21.4%)
Moderate effect on partners’ life	10/28 (35.7%)
Very large effect on partners’ life	9/28 (32.1%)
Extremely large effect on partners’ life	2/28 (7.1%)
FSFI-6 (score)	22.6 (2.1)
Sexual dysfunction in female partners (yes)	1/11 (9.1%)
IIEF-5 (score)	19.4 (1.4)
Erectile dysfunction in male partners (yes)	10/17 (58.8%)

Continuous variables are expressed as means (standard deviation) and qualitative variables as absolute (relative) frequencies. FDLQI: Family Dermatology Life Quality Index. FSFI-6: Female Sexual Function Index-6. IIEF-5: International Index of Erectile Function-5.

**Table 4 ijerph-20-00389-t004:** Factors associated with lower FDLQI, FSFI-6 and IIEF-5 scores in patients’ partners.

	FDLQI	FSFI-6	IIEF-5
Patient’s age	β = 0.13 (0.12), *p* = 0.2794	β = −0.15 (0.16), *p* = 0.3631	β = −0.25 (0.16), *p* = 0.1390
Partner’s age	β = 0.12 (0.13), *p* = 0.3535	β = −0.05 (0.18), *p* = 0.7902	β = −0.21 (0.16), *p* = 0.1891
Sex			
Male	$\bar{x}$ = 9.27 (2.16)	-	-
Female	$\bar{x}$ = 10.88 (1.74)		
	*p* = 0.5661		
BMI	β= 0.42 (0.2), ***p* = 0.0468**	β= 0.29 (0.34), *p* = 0.4164	β= −0.22 (0.23), *p* = 0.3511
Hurley stage			
I	$\bar{x}$ = 9.67 (2.54)	$\bar{x}$ = 24 (5.5)	$\bar{x}$ = 20.25 (2.9)
II	$\bar{x}$ = 7.88 (1.56)	$\bar{x}$ = 23.43 (2.94)	$\bar{x}$ = 20 (2.05)
III	$\bar{x}$ = 17.17 (2.54)	$\bar{x}$ = 16 (5.5)	$\bar{x}$ = 17.5 (2.9)
	***p* = 0.0161**	*p* = 0.4917	*p* = 0.7450
IHS4	β= 0.28 (0.13), ***p* = 0.0377**	β= −0.5 (0.42), *p* = 0.2671	β= −0.15 (0.12), *p* = 0.2164
Axilla			
Yes	$\bar{x}$ = 13.1 (2.17)	$\bar{x}$ = 24 (3.93)	$\bar{x}$ = 20.17 (2.32)
No	$\bar{x}$ = 8.67 (1.62) *p* = 0.1133	$\bar{x}$ = 21.14 (2.97) *p* = 0.5765	$\bar{x}$ = 19 (1.8) *p* = 0.6965

Breast			
Yes	$\bar{x}$ = 12.33 (4.14)	-	$\bar{x}$ = 20.33 (3.28)
No	$\bar{x}$ = 10 (1.43) *p* = 0.5986		$\bar{x}$ = 19.23 (1.58) *p* = 0.7666

Genitals			
Yes	$\bar{x}$ = 12.33 (2.35)	$\bar{x}$ = 23.33 (4.6)	$\bar{x}$ = 17.5 (2.24)
No	$\bar{x}$ = 9.26 (1.62) *p* = 0.2916	$\bar{x}$ = 21.75 (2.82) *p* = 0.7759	$\bar{x}$ = 20.6 (1.73) *p* = 0.2915

Buttocks			
Yes	$\bar{x}$ = 12.17 (2.91)	$\bar{x}$ = 24 (5.63)	$\bar{x}$ = 19.25 (2.85)
No	$\bar{x}$ = 9.73 (1.52) *p* = 0.4641	$\bar{x}$ = 21.78 (2.65) *p* = 0.7291	$\bar{x}$ = 19.5 (1.65) *p* = 0.9406

Groin			
Yes	$\bar{x}$ = 10.4 (1.61)	$\bar{x}$ = 21.57 (3.01)	$\bar{x}$ = 18.92 (1.62)
No	$\bar{x}$ = 9.88 (2.55) *p* = 0.8630	$\bar{x}$ = 23.25 (3.98) *p* = 0.7443	$\bar{x}$ = 21 (2.81) *p* = 0.5765

NRS pain	β = 0.59 (0.35), *p* = 0.1048	β = −0.67 (0.49), *p* = 0.1977	β = −0.57 (0.43), *p* = 0.2026
NRS suppuration	β = 0.52 (0.36), *p* = 0.1634	β = −0.85 (0.59), *p* = 0.1798	β = −0.25 (0.39), *p* = 0.5323
NRS unpleasant odor	β = 0.66 (0.3), ***p* = 0.0371**	β = −0.79 (0.5), *p* = 0.1471	β = −0.52 (0.31), *p* = 0.1150
NRS pruritus	β = 0.87 (0.32), ***p* = 0.0107**	β = −1.22 (0.6), *p* = 0.0719	β = −0.2 (0.35), *p* = 0.5821
Time of evolution	β = 0.11 (0.1), *p* = 0.2730	β = 0.06 (0.13), *p* = 0.6889	β = −0.29 (0.13), ***p* = 0.0437**

Variables are expressed as comparison of means (standard deviations) between categories when they are qualitative variables and as regression slope (standard deviation) when they are continuous variables. FDLQI: Family Dermatology Life Quality Index. FSFI-6: Female Sexual Function Index-6. IIEF-5: International Index of Erectile Function-5. BMI: body mass index. IHS4: International Hidradenitis Suppurativa Severity Score System. NRS: Numeric Rating Scale. Statistically significant *p* values are bold.

## Data Availability

Not applicable.

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
