# Peer review of "Sexual Dysfunction and Quality of Life in Patients with Hidradenitis Suppurativa and Their Partners"

_ijerph, 2022, doi:10.3390/ijerph20010389_

Round 1

Reviewer 1 Report

It is an interesting article on a subject on which there is little literature and very relevant for the quality of life of patients. It is a first study to consider the causes of sexual dysfunction in thse patients and their partners. It would be also very interesting to make a brief  reference to new studies that investigate the possible causes of sexual dysfunction, such as comorbidities and lifestyles. It is a study that includes few patients, but it opens the door to new investigations to better understand tha causes of sexual dysfunction.

Author Response

Thank you very much for your comments. Following your recommendations, we have made a brief reference to new studies that investigate the possible causes of sexual dysfunction, including comorbidities and lifestyles

Reviewer 2 Report

The topic of sexual impairment is of great importance in HS as it dramatically affects quality of life, especially when involves inguinal folds. Nevertheless, this topic has already been indagated in previous studies (involving more centers and more patients) but I noticed that most of them have not been added in the reference lists. Some of them:

  - Alavi A, Farzanfar D, Rogalska T, Lowes MA, Chavoshi S. Quality of life and sexual health in patients with hidradenitis suppurativa. Int J Womens Dermatol. 2018;4(2):74–79   - Sampogna F., Abeni D., Gieler U., Tomas-Aragones L., Lien L., Titeca G. Impairment of sexual life in 3485 dermatological outpatients from a multicentre study in 13 European countries. Acta Derm Venereol. 2017;97:478–482   - Kurek A., Peters E.M., Chanwangpong A., Sabat R., Sterry W., Schneider-Burrus S. Profound disturbances of sexual health in patients with acne inversa. J Am Acad Dermatol. 2012;67(3):422–428       Your study is well presented and has a scientific soundness. I understand that you indagated also sexual qol in partners but I think it is reasonable considering how dramatic is HS.  The role of inflammation could also be considered as HS presents with a remitting-relapsing course. This chronic behaviour can also condition qol and sexual impairment. For example in the introduction you listed some symptoms such as pruritus, odor and suppuration. In my opinion, pain is more important as disabling in daily activities including sexual intercourse. Pain is strictly related with inflammation, a parameter that can be evaluated with color doppler in ultrasound (Nazzaro G, et al. Color Doppler as a tool for correlating vascularization and pain in hidradenitis suppurativa lesions. Skin Res Technol. 2019 Nov;25(6):830-834) This is a suggestion for the limitation paragraph.  

Author Response

The topic of sexual impairment is of great importance in HS as it dramatically affects quality of life, especially when involves inguinal folds. Nevertheless, this topic has already been indagated in previous studies (involving more centers and more patients) but I noticed that most of them have not been added in the reference lists. Some of them:

  - Alavi A, Farzanfar D, Rogalska T, Lowes MA, Chavoshi S. Quality of life and sexual health in patients with hidradenitis suppurativa. Int J Womens Dermatol. 2018;4(2):74–79   - Sampogna F., Abeni D., Gieler U., Tomas-Aragones L., Lien L., Titeca G. Impairment of sexual life in 3485 dermatological outpatients from a multicentre study in 13 European countries. Acta Derm Venereol. 2017;97:478–482   - Kurek A., Peters E.M., Chanwangpong A., Sabat R., Sterry W., Schneider-Burrus S. Profound disturbances of sexual health in patients with acne inversa. J Am Acad Dermatol. 2012;67(3):422–428      

Thank you very much for all your comments. We have included all the references suggested in the introduction

Your study is well presented and has a scientific soundness. I understand that you indagated also sexual qol in partners but I think it is reasonable considering how dramatic is HS.  The role of inflammation could also be considered as HS presents with a remitting-relapsing course. This chronic behaviour can also condition qol and sexual impairment. For example in the introduction you listed some symptoms such as pruritus, odor and suppuration. In my opinion, pain is more important as disabling in daily activities including sexual intercourse. Pain is strictly related with inflammation, a parameter that can be evaluated with color doppler in ultrasound (Nazzaro G, et al. Color Doppler as a tool for correlating vascularization and pain in hidradenitis suppurativa lesions. Skin Res Technol. 2019 Nov;25(6):830-834) This is a suggestion for the limitation paragraph.  

We have also highlighted the importance of the pain in the introduction and the discussion as suggested.

Reviewer 3 Report

This is a well written paper on  an important subject, it certainly merits publication.

I have 2 minor remarks

Row 147 please specify the cut points, do you mean the score of 19 presented on the previous page

Table 2 HS (yes)  what does this mean?

Author Response

This is a well written paper on  an important subject, it certainly merits publication.

I have 2 minor remarks

Thank you very much for your comments.

Row 147 please specify the cut points, do you mean the score of 19 presented on the previous page

A score of 19 or less in FSFI-6 indicates sexual dysfunction in women and a score of 21 or less in IIEF-5 indicates ED. This sentence has been included in this line as recommended.

Table 2 HS (yes)  what does this mean?

It means the presence of sexual dysfunction in women with hidradenitis suppurative. The sentence has been centred to avoid confussion.

Reviewer 4 Report

Thank you very much for the opportunity to read this interesting manuscript entitled "Sexual dysfunction and quality of life in patients with hidradenitis suppurativa and their partners" by Carlos Cuenca-Barrales et al.

Hidradenitis suppurativa is a disease that affects intimacy, significantly reduces quality of life and can cause many psychological problems in patients such as depression and increased suicide rates.

This manuscript is very interesting, the study is very well organized, the results are presented clearly and the conclusions follow from the data presented. The exceptional uniqueness of the work is indicated by the multicenter (Spain and Poland).  The work is written in scientific language. I think that the work is very good, it deals with a very important problem. I noticed that the references are quite old. Only a few papers (5 out of 37 ) are from 2019-2021.

However, I have minor remarks, suggestions and comments, which are below : 

1) Introduction Section - I think , in a few sentences the authors should describe more extensively what HS is. Dermatologists are very familiar with the disease, but to make it more readable for non-dermatology readers I would suggest that the Introduction section be expanded to include the pathogenesis, clinical characteristics of the disease.  Please include these references : https://doi.org/10.5114/ada.2022.119008

Quality of life in HS is very important. 

Please consider new publication about QoL and Depression suggestions are below :   https://doi.org/10.5114/ada.2022.114885 https://doi.org/10.5114/ada.2022.119080  

Please consider this publication it’s very interesting about molecular aspects and clinical aspects of HS : https://doi.org/10.3390/cells10082094

2) Introduction Section - the authors actually do not mention the treatment of HS, in addition to dermatological treatment, surgical treatment is an effective method. This may underscore the importance of this manuscript, since patients in many cases are so desperate that they undergo extensive reconstructive surgery. He suggests mentioning surgical methods in the treatment of HS : https://doi.org/10.5114/ada.2022.115323

 https://doi.org/10.3390/jcm11092311

https://doi.org/10.1159/000462979

There are also completely new methods of surgical treatment of HS like co-graft acellular dermal matrix and split thickness skin graft : https://doi.org/10.3390/bioengineering9080389 

https://doi.org/10.3390/biomedicines10112870 

Please consider expanding these excerpts with these papers to emphasize even more the relevance of this work.

3) Discussion section : also suggests expanding the passage on surgical treatment and quality of life, actually health related life quality.  He suggests an interesting paper on assessing quality of life after surgical treatment : https://doi.org/10.3390/jcm11154327.

4) ISH4 was a median ? Please clarify. 

The English language is very good, no correction needed. The paper deserves its place in IJERPH because it deals with a very important aspect in HS. Thanks to this work also, perhaps more attention will be paid to the problem of HS patients in general. The work needs minor corrections, which are included in this review.

Author Response

Thank you very much for the opportunity to read this interesting manuscript entitled "Sexual dysfunction and quality of life in patients with hidradenitis suppurativa and their partners" by Carlos Cuenca-Barrales et al.

Hidradenitis suppurativa is a disease that affects intimacy, significantly reduces quality of life and can cause many psychological problems in patients such as depression and increased suicide rates.

This manuscript is very interesting, the study is very well organized, the results are presented clearly and the conclusions follow from the data presented. The exceptional uniqueness of the work is indicated by the multicenter (Spain and Poland).  The work is written in scientific language. I think that the work is very good, it deals with a very important problem. I noticed that the references are quite old. Only a few papers (5 out of 37 ) are from 2019-2021.

Thank you very much for all your comments. We have included more updated references as recommended.

However, I have minor remarks, suggestions and comments, which are below : 

1) Introduction Section - I think , in a few sentences the authors should describe more extensively what HS is. Dermatologists are very familiar with the disease, but to make it more readable for non-dermatology readers I would suggest that the Introduction section be expanded to include the pathogenesis, clinical characteristics of the disease.  Please include these references : https://doi.org/10.5114/ada.2022.119008

Quality of life in HS is very important. 

Please consider new publication about QoL and Depression suggestions are below :   https://doi.org/10.5114/ada.2022.114885 https://doi.org/10.5114/ada.2022.119080  

Please consider this publication it’s very interesting about molecular aspects and clinical aspects of HS : https://doi.org/10.3390/cells10082094

 We have expanded the introduction section and included the recommended references

2) Introduction Section - the authors actually do not mention the treatment of HS, in addition to dermatological treatment, surgical treatment is an effective method. This may underscore the importance of this manuscript, since patients in many cases are so desperate that they undergo extensive reconstructive surgery. He suggests mentioning surgical methods in the treatment of HS : https://doi.org/10.5114/ada.2022.115323

 https://doi.org/10.3390/jcm11092311

https://doi.org/10.1159/000462979

There are also completely new methods of surgical treatment of HS like co-graft acellular dermal matrix and split thickness skin graft : https://doi.org/10.3390/bioengineering9080389 

https://doi.org/10.3390/biomedicines10112870 

Please consider expanding these excerpts with these papers to emphasize even more the relevance of this work.

We have included some sentences about HS management and cited your suggestion 

3) Discussion section : also suggests expanding the passage on surgical treatment and quality of life, actually health related life quality.  He suggests an interesting paper on assessing quality of life after surgical treatment : https://doi.org/10.3390/jcm11154327.

We have included this citation in the introduction

4) ISH4 was a median ? Please clarify. 

It is a continuous variable so it was expressed as means (standard deviation) as mentioned in material and methods

The English language is very good, no correction needed. The paper deserves its place in IJERPH because it deals with a very important aspect in HS. Thanks to this work also, perhaps more attention will be paid to the problem of HS patients in general. The work needs minor corrections, which are included in this review.

Thank you for the comments

Round 2

Reviewer 4 Report

Authors well adressed all comments and suggestions. 

I’m strongly for accept this interesting paper